# A Food Relief Charter for South Australia—Towards a Shared Vision for Pathways Out of Food Insecurity

**DOI:** 10.3390/ijerph19127080

**Published:** 2022-06-09

**Authors:** Tahna L. Pettman, Carmel Williams, Sue Booth, Deborah Wildgoose, Christina M. Pollard, John Coveney, Julie-Anne McWhinnie, Marian McAllister, Carolyn Dent, Rory Spreckley, Jonathan D. Buckley, Svetlana Bogomolova, Ian Goodwin-Smith

**Affiliations:** 1Centre for Social Impact Flinders, College of Business Government and Law, Flinders University, Adelaide, SA 5000, Australia; svetlana.bogomolova@flinders.edu.au (S.B.); ian.goodwinsmith@flinders.edu.au (I.G.-S.); 2Wellbeing SA, Government of South Australia, Adelaide, SA 5000, Australia; carmel.williams@healthtranslationsa.org.au (C.W.); deborah.wildgoose2@sa.gov.au (D.W.); julie-anne.mcwhinnie@sa.gov.au (J.-A.M.); 3College of Medicine and Public Health, Flinders University, Adelaide, SA 5000, Australia; sue.booth@flinders.edu.au; 4School of Population Health, Curtin University, Perth, WA 6102, Australia; c.pollard@curtin.edu.au; 5College of Nursing and Health Sciences, Flinders University, Adelaide, SA 5000, Australia; john.coveney@flinders.edu.au (J.C.); marian.mcallister@flinders.edu.au (M.M.); carolyn.dent@flinders.edu.au (C.D.); 6Department of Human Services, Government of South Australia, Adelaide, SA 5000, Australia; rory.spreckley@sa.gov.au; 7Allied Health and Human Performance, University of South Australia, Adelaide, SA 5000, Australia; jon.buckley@unisa.edu.au

**Keywords:** food assistance, food relief, food insecurity, policy, intersectoral collaboration, collective impact, co-production

## Abstract

Chronic food insecurity persists in high-income countries, leading to an entrenched need for food relief. In Australia, food relief services primarily focus on providing food to meet immediate need. To date, there has been few examples of a vision in the sector towards client outcomes and pathways out of food insecurity. In 2016, the South Australian Government commissioned research and community sector engagement to identify potential policy actions to address food insecurity. This article describes the process of developing a co-designed South Australian Food Relief Charter, through policy–research–practice collaboration, and reflects on the role of the Charter as both a policy tool and a declaration of a shared vision. Methods used to develop the Charter, and resulting guiding principles, are discussed. This article reflects on the intentions of the Charter and suggests how its guiding principles may be used to guide collective actions for system improvement. Whilst a Charter alone may be insufficient to create an integrated food relief system that goes beyond the provision of food, it is a useful first step in enabling a culture where the sector can have a unified voice to advocate for the prevention of food insecurity.

## 1. Introduction

Despite strong economies and an abundant food supply in high-income countries, individual and household chronic food insecurity persists due to poverty and inadequate levels of social protection [1]. Food insecurity (a lack of access to safe, sufficient, and appropriate food [2]) is a complex social problem that is represented and understood in different ways [3,4]. Previously represented as a problem of individual or community responsibility that deflects from the need for government intervention [1,4], the determinants of food insecurity are increasingly understood to result from inadequate social protection, low wage growth, a lack of wage-setting policies, unemployment, and homelessness [1,5,6,7].

While population prevalence of food insecurity is not routinely measured in Australia [8,9], it has been estimated to be approximately 4% [10,11] and higher in several population sub-groups due to inequities [8,12,13,14]. In South Australia (SA), state government surveillance in 2020 identified that at least 8.5% of adults reported experiencing food insecurity in the past 12 months [15]. Since 2020, the COVID-19 pandemic containment measures, economic impacts, and food supply disruptions have exacerbated food insecurity for some Australians, particularly following the withdrawal of economic support packages [16,17,18,19,20]. 

### 1.1. Challenges in the Food Relief Sector

Charitable food relief is the dominant response to food insecurity in Australia, which mostly relies on donations from the not-for-profit sector with support from state and federal government grants, donors, and partnerships with supermarkets and food manufacturers [21]. The food relief sector includes numerous ‘direct’ services that provide food relief to people (food banks, food pantries, soup kitchens, and other community organisations) [22], as well as some ‘indirect’ services (larger organisations that rescue or bank and redistribute food to direct services) [22]. Food rescue/redistribution organisations may receive some national Government funding and provide food to direct services at a cost (handling fees). Various direct service types exist including food parcels/hampers, pantries, food vans, supermarket vouchers, and seated meal services [22], and emerging not-for-profit social enterprise models and community initiatives such as social supermarkets [23,24]. The sector primarily relies on charitable donations, a volunteer workforce, and variable funding sources, and foods available are largely determined by supermarket donations and the food redistribution chain. This often results in a reliance on shelf-stable foods that are of insufficient nutrient quality [22,25]. Few food relief agencies have nutrition standards in place [22,25,26,27].

The food relief sector reports that the demand for food relief has been persistent in recent years [20,28,29,30]. Government surveys in 2020 suggest approximately 38% of families reported at least one indicator of financial distress, and of these, 4% of families sought assistance from welfare/community organisations (including food relief), which had doubled from the previous survey period [31]. Factors such as unemployment and insecure work, unpredictable income from jobs in the gig economy, increased private rents, and insecure housing also contribute to demand [32,33]. People often seek help from friends and families first, and use food relief through welfare/community sector organisations as a last resort [34]. As such, reported figures of access to food relief may understate the actual need.

To date, the ad hoc collection and redistribution of food, and direct food relief, has not yet translated to reduced food insecurity [1,22,30]. The wide range of service providers operate independently as not-for-profit (or ‘for purpose’) organisations, and the sector has not been orientated towards the goal of ending food insecurity through service coordination or a focus on underlying determinants. Some agencies provide case management, referrals, and social supports, but the sector is not resourced well to do this work [30], as the scope of Government funding generally only allows the provision of food for ‘emergency assistance’. In line with this way of procuring services, such that the focus is on discrete support to meet an immediate need (i.e., distribute food) rather than achieving broader social outcomes (i.e., reduce food insecurity), no mechanisms are in place to support or systematise sector coordination, accountability, or governance. This has resulted in a self-organised sector and a continued proliferation of relatively simple approaches (distributing food, education) to address a complex problem (poverty, social exclusion, etc.) [6,22,30,35,36,37]. 

Despite this, food relief remains a significant part of the Australian welfare response, and it is critical to ensure that these services are supported to work towards an optimal system, ensuring quality food and dignified service to those who are most vulnerable in society. Research continues to report experiences of limited choice, poor food quality, shame, stigma, and embarrassment [34,38,39]. To better address food insecurity, it is necessary to view the system (sector) as more than the sum of its parts (services). The creation of a shared vision, such as a Charter for all food relief services, is one policy tool that could help to set the direction for change across the system. A united voice across the many diverse services that make up the sector could be used to generate commitment and momentum towards practice improvement and orient efforts towards client outcomes. 

### 1.2. Charters as a Policy Tool 

Charters are a policy tool that may function as a governing document for an individual organisation, or group of organisations. Charters may be a statement of purpose and governing structure of commitment, roles, and responsibilities [40,41], of rights [42,43], or of a commitment to shared values within or between organisations [44]. Service charters are also common in service settings to express commitment to provide quality service [45], set out standards of service that clients can expect, clients’ rights, and processes for complaints [46,47]. Service charters aim to ensure that organisations focus on service delivery, performance measurement and improvement. Charters are used to document agreed visions and goals, and improve service quality and user satisfaction [48]. As a policy tool, many of these examples of Charters appear symbolic in their intent, providing aspirational principles and values for signatories to work towards. 

In Western Australia (WA), a framework [49] similar to a Charter was initiated by the social services sector due to an interest in coordinating the disparate sector and improving population outcomes. The WA Food Relief Framework aimed to use an outcomes-oriented service delivery to promote flexible services tailored to the needs and circumstances of the clients and develop Practice Principles for Community Relief and Resilience [49]. 

Charters or similar instruments are rarely used in the Australian food relief sector and have not been initiated by Government in any other Australian jurisdiction. 

This article examines the co-development of an SA government-led Charter, as one part of an overarching intersectoral collaboration project. The Charter aims to provide an aspirational vision and set of guiding principles for the food relief sector, to work towards an optimal food relief system. This article considers the policy context, intentions of the Charter, and elaborates on how its principles could be used to guide collective efforts for system improvement and client outcomes.

## 2. Materials and Methods

### 2.1. The Stages of Co-Development of the South Australian Food Relief Charter 

#### 2.1.1. Pre-Charter Development (Stage 0): Policy Initiation, Intersectoral Collaboration, and Community Sector Engagement

Acknowledging a need to improve the health, wellbeing, and social outcomes of South Australian communities, in 2015, the state government Department of Health and Wellbeing invited the Department of Human Services to work in partnership across Government. A ‘Public Health Partner Authority’ (PHPA) agreement [50] provided the authorising mechanism that allowed the two departments to progress collaborative action. Food security was identified as one of the PHPA’s initial key focus areas. SA, through its longstanding Health in All Policies (HiAP) initiative [51], had a strong background in partnership and co-design approaches, which provided a supportive policy environment and resourcing for a collaborative workplan [52]. The partnership is an ongoing initiative which “seeks to improve food security and health and wellbeing outcomes through strategies and actions to implement the recommendations of the *Improving Individual and Household Food Security Outcomes in SA”* report [53]. 

Drawing on the HiAP experience [51], policy actors from the two State Government agencies commissioned research and commenced a process of food relief sector engagement to address food insecurity through collaboration across sectors. For the purpose of this paper, ‘policy actors’ refer to the various individuals in public servant roles in State government. While these individuals and their agencies are not often able to influence other actors’/agencies’ strategies [54], the PHPA provided an authorising environment to set a joint workplan with goals of a shared interest to both agencies: improving individual and household food security. The HiAP process that was followed includes evidence gathering, identifying solutions, generating policy recommendations jointly owned by stakeholders, then navigating recommendations through decision-making processes [52]. A policy–research–practice collaboration was established to progress the food security partnership workplan. The process and outputs of this collaboration are graphically summarised in Figure 1.

A critical part of this formative stage was planning for community sector engagement in the evidence-gathering process, prior to solutions-generation. Given the inherent limitations of the voluntary sector [28], and in recognition of their role as the principal mechanism for the provision of food relief in Australia, extensive collaboration between government and the non-profit sector emerges as the “logical and theoretically sensible compromise” ([55], pp. 42–43). 

The formative evidence-gathering and solutions-generation phase of the food security partnership included scoping research and a review of existing evidence (October 2016) [56], sector engagement and consultations via roundtable discussion (August 2017) [57], a state-wide food relief provider survey (September 2017) [58], and a discussion paper for community sector consultation (November 2017). A research project that gathered insights for service improvement from the perspectives of food relief recipients [59] (November 2017), and a second consultation via roundtable discussion with the food relief sector (December 2017) [60] were used to develop draft recommendations. 

#### 2.1.2. Stage 1: Initiation of Charter Project

Following the previous two years of comprehensive consultation and sector engagement, the evidence and draft recommendations were incorporated into a summary report released in February 2018 (*The Improving Individual and Household Food Security Project final report* [53]). In this report, two of the key recommendations identified the need to develop a shared ‘best practice framework’ and set of agreed outcomes for the sector. This reflected the sector’s recognition that a shared vision, guiding principles for practice, and outcome indicators were needed if the sector was to effectively tackle the current challenge. Four of the recommendations were identified for initial implementation (Figure 1). This article describes implementation of recommendation 1.2 ‘A best practice framework for the provision of food relief’. Two other recommendations, 2.3 and 3.1, have progressed separately and are not the focus of the current article. The process of development of the SA Food Relief Charter is illustrated in Figure 2. 

Researchers with experience in community development, public health, social service, food insecurity, and food systems (University of South Australia and Flinders University) were engaged to partner in the process with the community sector (food relief organisations). Importantly, this relationship identified a food relief framework developed in WA following an extensive consultation of the sectors in that state [49]. A copy of the WA framework, ‘Consumer and Provider Charter’ published in 2019 [49], was provided to the South Australian project team. This provided valuable insights into planning of a similar process and principles relevant to SA.

To initiate the process of co-developing a ‘best practice framework’, a small working group of government policy actors and researchers was formed to work in consultation with community sector organisation practitioners. Initially this involved a brief desktop review of peer-reviewed and grey literature exploring examples of sector-relevant Charters, and inductive engagement with the food relief sector through interactive workshops. This was augmented with personal communication with food security experts and academics in other states of Australia, to identify unpublished documents of relevance (including the WA framework). 

Following the desktop review and extraction of common relevant principles, a Charter (formerly ‘best practice framework’) was chosen as the policy tool to embody a set of principles to guide ‘best practice’ service delivery. It was intended that food relief agencies would co-create the Charter and become signatories to express their commitment to the principles and desired outcomes.

#### 2.1.3. Stage 2: Co-Development of a ‘Best Practice Framework’ (the Charter) with the Sector

The research team facilitated the engagement and co-design process with the sector. The aim was to develop an agreed set of guiding principles that would set out a vision and some directions for progression across the sector, and to develop an SA Charter—a government-led policy tool that acknowledged sector stakeholders’ commitment to progressing the principles. A set of draft principles were developed based on this work as a starting point for an SA Charter. Client voice was represented in the principles through explicit incorporation of recommendations from the SA formative research [59].

In April 2019, a workshop was convened with food relief sector representatives. Participants provided feedback on the early draft principles, firstly in small group discussion, after which suggestions were brought back to the wider group for agreement. The final principles were drafted based on this feedback, with input from the policy actors in the two Government agencies. Between May–June 2019, individual consultations were held with community sector organisations via phone/email to seek feedback and responses to the draft revised principles.

#### 2.1.4. Stage 3: SA Food Relief Charter Completion and Launch with Signatories

Sector organisation responses were incorporated to form a set of principles and outcomes-focused commitment statements. The content and description of the Charter launch are described in the following section. 

## 3. Results

The processes described (initiation, desktop research, sector engagement, co-development, signing, and launch) resulted in a shared commitment among all signatories and a set of five guiding principles to guide future actions across the food relief sector.

### 3.1. Guiding Principles

The most common and relevant best practice human service principles and elements drawn from the document review that aligned with promoting positive outcomes for people and communities are summarised in Table 1. The existing Charters examined included principles with explanatory sentences and a range of elements pertaining directly to individuals’ access to food in the food relief sector. Best practice principles were also identified in broader vision and mission statements, and in service standards and improvement goals. These existing practice principles and their related key words and phrases were synthesised to document a list of draft principles for the South Australian context. 

The synthesis of available information led to a draft list of Principles that could be discussed and deliberated by the community sector. In the first consultation workshop (April 2019), the draft list of principles was offered as a potential basis of a Charter. Post-Workshop, a second draft was developed, taking into consideration the sector feedback as well as input from policy actors. Further consultation and individual feedback was sought from the sector by email and or telephone (May–June 2019). The resulting Guiding Principles for the SA Food Relief Charter are summarised in Table 2, with accompanying explanatory statements (full version in the Appendix A).

### 3.2. Goals—Commitment Statements

As well as having outcomes-focused statements that accompanied the principles, four ‘commitment of partners’ statements were defined in the Charter. These ‘commitment of partners’ statements were generated during the refinement of the guiding principles, and draw on earlier work [57]. These were intended to serve as a reminder to the sector of an ongoing need to work in partnership to move beyond outputs, to client and community outcomes, including:Increased food and nutrition security for South Australians and improved health outcomes;Build sector-wide standards for best practice in the food relief system;Create opportunities for people to build skills and have the capacity to move out of food insecurity; andBuild a skilled and sustainable food relief sector workforce.

### 3.3. Charter Signatories

The signing of the Charter symbolised the joint goal of the government and the community sector to commit to work collaboratively towards an optimal food relief system that leads to improved client and community outcomes. In November 2019, the South Australian Minister for Health and Wellbeing and the Minister for Human Services co-launched the SA Food Relief Charter [64]. At the launch, 15 community sector organisations signed the Charter. Signatories included representatives from food redistribution/rescue agencies, direct food relief providers, associations/advocacy organisations for community service agencies, and community members in both metropolitan and regional locations. Additional agencies have been invited by the Department of Human Services to sign the voluntary Charter as they commence new food relief funding agreements. 

## 4. Discussion

This article documents the collaborative process to bring together two government agencies, academic researchers, and the food relief sector to co-develop a policy tool and a declaration of a shared vision. The Charter documents expressed commitment to common goals of collaboration for a collective impact, quality service, and improving client outcomes. To our knowledge, there are no other government-led Charters in Australia developed with and for the food relief sector. The WA framework is the first Australian example of a similar tool that recognises that improvements to the food relief service system can be made to better respond to need, and that provides a roadmap to improved client outcomes [49]. ‘More Than Food’ from the USA is another framework with similar principles, in that it focuses on building capacity in food relief services to more effectively address the underlying causes of food insecurity [65].

While the SA Charter is not a governance document or service charter in the true sense, it offers an explicit policy tool by which signatory agencies can express commitment to the shared values, goals, and principles for sector improvement [49]. This is consistent with international practice where Charters are used to document a shared vision or goal and set directions for the signatories [44,45,46,49]. While the SA Charter does not link to service delivery responsibilities or accountability reporting, and is not intended as an implementation instrument, the Charter documents a shared vision from the community sector, and may help to begin conversations across the sector about quality improvement and client outcomes. The process of Charter co-development was a critical foundation to build relationships and trust within the community sector, which may enable future co-design work around the adoption and translation of the Charter Principles into practice.

### Strengths and Limitations

The Charter development process has a number of strengths: First, and of most importance in terms of contribution to knowledge and practice, the documentation of this process offers a template for future policy actors and practitioners to follow, having meticulously documented all of the stages to co-design the Charter. The process was deliberatively inclusive of multiple stakeholders and actors, sources of evidence, and included insights from research with clients. This is comparable to other Australian states’ food security strategies and principles [49,66,67]. While this process of engagement and co-design is not new, what is unique in the Australian context is the development of a Charter by three distinct sectors (Government, community/for-purpose, and academia). Furthermore, for the first time, the process resulted in the acknowledgement and agreement of a vision and specific principles for improved practice, outcomes, and monitoring. These commitments focused on client outcomes and pathways out of food insecurity, and represent a real step-change from the provision of a few days’ emergency food relief, which has been the stated purpose of food relief since its inception.

Second, a unique aspect of this work was the leadership from state government and its efforts to maintain a long-term close collaboration between policy actors across two sectors, academic researchers, and critically, the community sector. This ‘joined up’ approach is increasingly promoted as an effective approach to policy making. Strengths of the process include stakeholder engagement at policy inception and co-design throughout formulation to maximize the buy-in, which may create ownership among participants/contributors [68,69]. Long-term resourcing of the partnership’s work, policy actors’ championing of collaboration, and power-sharing with community practitioners are also acknowledged, which may have contributed to the Charter’s legitimacy and the ongoing commitment of all parties. These outcomes are noteworthy considering the changes in governance that were naturally experienced throughout the duration of the project [70]. 

Third, this article contributes to evidence about knowledge co-production. While many academic institutions, funders, and governing bodies strive to document impacts beyond academia, it remains challenging to claim a demonstrable influence upon policy, practice, or the lived experience of individuals. Various factors affect the extent to which engagement between researchers and policy actors can be instrumental and effective [71]. In this case, the early involvement of researchers in evidence gathering and policy development meant that the policy tool was informed by research, theory, as well as the inductive evidence and insights of clients. 

A challenge for researchers through the project was in moderating an idealised expectation for the policy tool, for example, one that more strongly oriented the sector towards addressing underlying determinants and shifting practice from outputs of providing food, to longer-term outcomes. For the food relief sector, largely charity-funded volunteer-run organisations, limited resources meant that agreeing to the Charter’s principles and aspirations challenged their service/business models and priorities.

A great opportunity exists to improve innovation, integration, and collaboration across the food security sector in Australia [72], and sector agencies unilaterally agree that as an outcome, all people should be food secure [56,67,73]. There are growing calls to transition away from ‘emergency assistance’ to more dignified, inclusive approaches, such as social supermarkets [23,28] and community-based initiatives [67]. Whilst the Charter alone may be insufficient to create an optimal food relief system, it is a useful first step in enabling a culture where the sector can have a unified voice, to advocate for the prevention of food insecurity. At a local level, the potential value of the Charter is in its documentation of a shared commitment to more sophisticated solutions to address food insecurity. It sets out shared goals for sector improvement (food quality, practice standards, client outcomes, workforce), and defines guiding principles for achieving this. The Charter is underpinned by elements of collective impact theory [74,75] in that it is driven by a common agenda based on a clearly articulated and common understanding of the issues. 

The principles provide guidance for collaborative practice across diverse services, which aim to guide mutually-reinforcing actions:

The first principle on service coordination and integration calls for a focus on systems, rather than services. Increased coordination and integration across different agency types, resources, operating times, service offerings, and geographic coverages could provide better client experiences and better links between redistributors, direct food relief providers, and community social enterprise models [28,72]. One example implemented in the US includes partnerships and co-locations between community-based food banks and healthcare settings [76]. A ‘coordination’ focus allows for a broad ecosystem of components working in harmony, without prescribing a single service model. 

The second principle encourages the use of nutrition guidelines to improve the food supply quality for vulnerable populations, complementing efforts to ensure choice and retail-like environments [37,77,78]. Embedding nutrition standards may help to limit the disproportionate supply of energy-dense nutrient-poor quality food and drinks from the redistribution supply chain [6]. 

The third principle focuses on ensuring dignified experiences and a values-based service culture. The Charter acknowledges the critical importance of a respectful, quality, and dignified service, which is consistent with recommendations from recent research with food relief clients [34,38,39]. The fourth principle emphasises utilising the ‘opportunity’ of food relief to engage people in wraparound services, social and skill development opportunities, commensality (‘social eating’/food sharing initiatives) [28,59,79,80], and to foster social inclusion. This is important because the need for food relief is often a symptom of poverty and social exclusion [72,81,82]. Multiple points of social connection are possible through commensality and are of value to people who are otherwise excluded from these experiences [83]. The Charter signposts the opportunity for food relief services to contribute to more progressive models such as social supermarkets [23], or less ‘bricks and mortar’ alliances of services orientated towards similar design principles. Hybrid social food initiatives (such as cafes, shared meals, and community cooking) can be a point where food relief and ‘eating out’ intersect [83], as there is a value to the social dimensions of eating and food. 

Finally, the fifth principle sets out a need to work towards the measurement of shared outcomes across the sector. A greater focus on data collection is overdue, and food relief agencies could be better supported to collect and share information in order to understand the social and health impacts of their activities. An evaluation of the relative contribution of various food relief models and their activities to meeting shared sector-wide goals would provide better knowledge to inform practice and policy decision making. 

While new work is commencing to understand the SA Charter principles’ translation into service practice, and no measurement of use or impact is available, anecdotal accounts have emerged that demonstrate a positive enactment since its release in 2019. For example, new collaborations have emerged between agencies in the sector (principle 1), especially around food distribution. Some agencies have increased their choice of groceries (principle 2) and have reduced eligibility assessments for access, to improve dignity. Nutritional quality is considered more often, with increased on-site assessments of the foods being accepted and some providers starting to augment food provision with purposely-sourced (even purchased), healthier foods to balance out donations (principle 2). Wraparound social services and opportunities for commensality are being explored in some agencies through on-site offerings or partner referrals (principle 4). 

The Charter shows promise in articulating a vision to improve the lived experience and outcomes of those receiving food relief, provided that services have enough food and organisational support to enact it. The impact of the overall food relief system on the reduction of food insecurity could be improved by more comprehensive routine monitoring. Government assistance payments and processes remain sub-optimal in Australia, despite social services advocating for increases in a livable income and basic wages. Food supply impacts and the high cost of living are also likely to increase demand for food relief in Australia, as is occurring in other countries [76]. For all these reasons, it is important for Australia not to follow the lead of countries providing inadequate social protection and relying on charitable food relief as the primary response [1]. 

## 5. Conclusions

Chronic reliance on food relief is an ongoing social and public health challenge which requires more sophisticated solutions. A government-initiated process brought together stakeholders from across the food relief sector to identify and commit to a shared vision for the sector. Collaboration between the government, academia, and community agencies led to a co-designed Charter documenting a shared goal and guiding principles to stimulate action in pursuit of that goal. The Charter sets out principles to encourage services to ensure that the experience of accessing food relief is an opportunity for commensality and social inclusion. As a policy tool, the Charter is a step towards an integrated optimal food relief system that goes beyond the provision of food.

## Figures and Tables

**Figure 1 ijerph-19-07080-f001:**
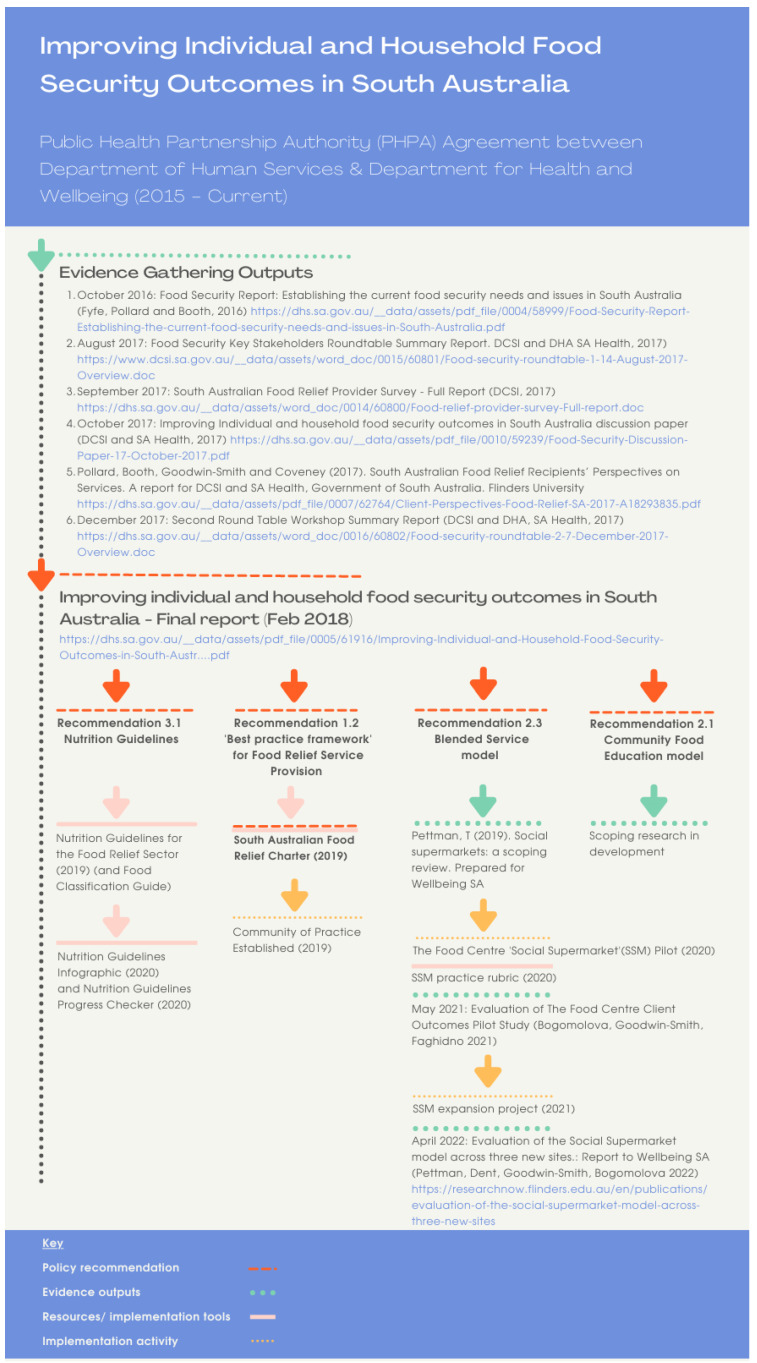
Flowchart of outputs of the overarching intersectoral collaboration—food security partnership.

**Figure 2 ijerph-19-07080-f002:**
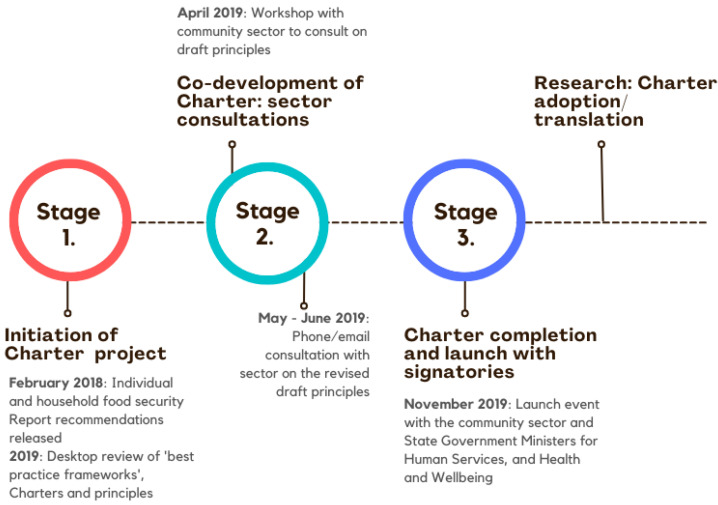
Process of the SA Food Relief Charter co-development.

**Table 1 ijerph-19-07080-t001:** Summary of guiding principles drawn from existing Charters/frameworks relevant to food relief service.

Principle	Key Words/Phrases about Each Principle	Source
Access and Equity, Collaboration	Maximise access and ensure equity in the delivery of services Provide quality services which are effective, efficient, and appropriately targeted Provide a dignified, welcoming, safe, and accessible environment Ensure assistance is available throughout the year Work collaboratively in delivering services and liaise with other organisations for the benefit of clients Refer clients to support services in the community to help address the underlying causes of financial crisis Establish or participate in community networks to facilitate tailored, coordinated service delivery across organisations Commit to and participate in relevant training opportunities and ensure paid workers, volunteers, and training facilitators are appropriately qualified	The Good Food Principles [61]; ACOSS Emergency Relief Handbook—Guiding Principles and Service Standards [62]
Nutrition focused	‘Believing and investing in the power of good food’ Serving unhealthy food is no longer good enough Responding to dietary needs and preferences that consumers are entitled to Sourcing appropriate foods for consumer needs and preferences	Beyond the emergency: How to evolve your food bank into a force for change [61]; WA Food Relief Framework—Consumer and Provider Charter [49]
Respect and Dignity	Respect and community leadership: We believe that respect should underpin all of our work. Thus we strive to avoid the signs, symbols, and procedures that contribute to the stigma often experienced by people attending food programs in charitable organisations, and to positively communicate our respect for all participants through respectful procedures and inviting community involvement Approaching the problem from the individual level to the systemic level: we believe that all people have the right to the basics of a dignified life—a decent income, housing, and employment Dignity at the root of services Ways to empower and encourage community voices and participation ‘Tailored and Respectful’ service Client choice Service is dignified No stigma in seeking assistance	Beyond the emergency: How to evolve your food bank into a force for change [61]; WA Food Relief Framework [49]
Person-centred, connections	‘Meeting people where they are at’—meet needs relevant to their circumstances ‘Taking action from the individual to the systemic’: food access, food skills and civic engagement; help meet basic need in the short term and maximise the choices available with skills that enable them to choose, grow, and prepare good food; offering programs that span the range of access, skills, and engagement on food and hunger creates relevance and multiple points of connections Consumer involvement and client participation in the formulation of procedures	The Good Food Principles [61]; The Salvation Army Doorways Handbook [63]
Impact measurement	Mechanisms exist to quantify and qualify outputs and outcomes on an ongoing basis, so the value of the service is always known ‘Aiming high for our organisation and our community’: [food relief] organisations need to be properly resourced and staffed to create impact ‘We hold ourselves to a high standard of performance and impact’	WAFRF—Practice Principles for Community Relief and Resilience [49]; The Good Food Principles [61]

**Table 2 ijerph-19-07080-t002:** Summary of the Guiding Principles of the South Australian Food Relief Charter.

Number	Principle	Relevant Desired/Intended Outcome
Principle 1:	Collaboration to build an effective and integrated food relief system	Improved service coordination in the food relief ‘system’ to improve impact, reducing the number of people reliant on food relief
Principle 2:	Focusing on nutrition and health	Maximise availability of healthy and appropriate foods, minimise provision of unhealthy food and drinks
Principle 3:	Delivering a service built on fairness and equity	Service based on values including choice, safety, dignity, respect, compassion, transparency, privacy, cultural sensitivity, empowerment, and independence, with a focus on action to assist people to move out of food insecurity
Principle 4:	Connecting people, building skills, and confidence	Accessing food relief provides an opportunity for engagement with other services and may facilitate pathways out of food insecurity
Principle 5:	Monitoring and evaluating to measure collective impact	Data collection to quantify and assess the quality of outputs and outcomes, with the view to develop a set of shared outcomes in future

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
