# Peer review of "A Food Relief Charter for South Australia—Towards a Shared Vision for Pathways Out of Food Insecurity"

_ijerph, 2022, doi:10.3390/ijerph19127080_

Round 1
Reviewer 1 Report
It was a pleasure to read this paper. The topic is an interesting one and the paper is well written. Every stage of co-development of the South Australian Food Relief Charter is well described. Answers to some questions were addressed in Strengths and Limitations section.
The paper demonstrate an adequate understanding of the relevant literature in the field. Therefore, I recommend this paper for publication.
Best regards.
Author Response
Reviewer 1 |
|
It was a pleasure to read this paper. The topic is an interesting one and the paper is well written. Every stage of co-development of the South Australian Food Relief Charter is well described. Answers to some questions were addressed in Strengths and Limitations section |
We appreciate the reviewer acknowledging the strengths of our manuscript. |
The paper demonstrate an adequate understanding of the relevant literature in the field. Therefore, I recommend this paper for publication |
Noted – thank you! |
Reviewer 2 Report
This manuscript presented a process of how the research team developed a charter to address the food insecurity issue in South Australia.
I have my concerns in the following two aspects:
One is that a Charter was developed by government departments and a research team that targeted to set a vision for a group of charity organizations that largely rely on volunteers. It seems a charter in this case may do more harm than good to these charity organizations in that a charter may serve as another red tape, considering this charter was not initiated by some of those charity organizations at all. There is a lack of convincing arguments in why a chart will help in addressing the food security issue in this case.
As stated by the authors that “The sector primarily relies on charitable donations, a volunteer workforce, variable funding sources, and foods available are largely determined by supermarket donations and the food redistribution chain. This often results in a reliance on shelf-stable foods that are of insufficient nutrient quality.”
Another aspect is that the charter development process is so common, mainly desktop research plus workshops with only some stakeholders (unfortunately charity organizations were not well engaged), really nothing new.
If authors reported the impacts of the developed charter on the food insecurity issue, it would be a very meaningful manuscript. I would think this manuscript is not mature yet.
Author Response
Reviewer 2 |
|
This manuscript presented a process of how the research team developed a charter to address the food insecurity issue in South Australia I have my concerns in the following two aspects One is that a Charter was developed by government departments and a research team that targeted to set a vision for a group of charity organizations that largely rely on volunteers. It seems a charter in this case may do more harm than good to these charity organizations in that a charter may serve as another red tape, considering this charter was not initiated by some of those charity organizations at all. |
In response to your concerns regarding the potential harm that the Charter could confer, we offer the following: Firstly, the Charter was not developed by government departments and a research team alone, it was co-designed with the sector and the initial recommendation to do a Charter came out of the consultation process with the sector. The consultation process was engaged multiple organisations within the sector, the outcome of which led to the idea to form a Charter. The Charter was an answer to the sector’s call for a set of guiding principles for best practice, something they can use to guide the development of their own processes. Thirdly, the Charter was developed with Salamon (1987) theory of government - non-profit relations in the welfare state in mind, particularly of the inherent limitations in both the private market and government as providers of collective goods. Given the inherent limitations of the voluntary sector and in recognition of their role as the principal mechanism for the provision of food relief, extensive collaboration between government and the non-profit sector emerges as the “logical and theoretically sensible compromise”. We have added a more specific description to the manuscript to clarify the initiation of the Charter and role of community sector organisations in its inception, in lines: 129-132, 135 – 139, 143, 148 – 149, 167. We have also added in text to address concerns regarding government - non-profit relations in the welfare state line 129 - 132 We have also added a process diagram to improve clarity, figure 2 (as suggested by reviewer #4) |
There is a lack of convincing arguments in why a chart will help in addressing the food security issue in this case. |
We agree that the Charter in and of itself is only one mechanism designed by the sector that offers principles that may be used to guide collective actions for system improvements to address food security. But it is a first and a necessary step to develop a shared vision of how the entire sector and its every member can shift their practices to ensure their service delivery promotes sustained exit from food insecurity for clients, not just perpetuate the reliance on limited resources. This aspirational Charter, if followed, should assist the sector to collaborate, focus on and engage strategies that support people to break the cycle of dependency on food relief. Essentially the Charter is a ‘best practice framework’ to guide practice in service provision. We have added clarity to the manuscript in lines: 139 - 141
|
Another aspect is that the charter development process is so common, mainly desktop research plus workshops with only some stakeholders (unfortunately charity organizations were not well engaged), really nothing new |
Yes, we acknowledge the process of development is not new. Yet, Charters provide a valuable instrument to guide actions, on governance, partnerships and sector development. According to our formative review of empirical evidence and grey literature, Charters co-developed with the charitable food relief sector remain relatively new in the Australian context, and, they offer a mechanism for ongoing sector relationship-building, service improvement and reform. While not a novel process, it was supported by the engagement with community organizations was done multiple times over two+ years. What is new is the development of the Charter by the three distinct sectors in the Australian context (government, NFP and academia), and for the first time, the acknowledgement and agreement of specific principles improved collaboration, nutrition focus, equity, service connections to facilitate pathways out of food insecurity, and a commitment to monitoring and evaluation. These commitment aim to ensure client outcomes and pathways out of food insecurity, a real change from the provision of 2-3 days emergency food relief that has been to stated purpose since its inception. Clarity has been added in line 240 - 245 with the previous two comments, to be clearer about the initiation of the Charter and role of community sector organisations in it’s inception. |
If authors reported the impacts of the developed charter on the food insecurity issue, it would be a very meaningful manuscript. I would think this manuscript is not mature yet |
Thank you for the suggestion. Pleasingly, this is precisely what we are about to work on with multiple community sector agencies, having recently received new partnership research funding to do so. While the charter is not an implementation tool in it’s present form, out of respect for the community sector partners, our new research with the sector will explore how the Charter has been perceived and adopted since it’s release in 2019.
Because the purpose and key message of the current manuscript is to describe the development of the Charter, we are not in a position and it is beyond scope to describe implementation and impact at this stage. We have attempted to communicate that the development process is a critical first step/foundation for trust-building with the community sector. This has now enabled us to do co-design work starting with three community sector organisations later this year.
We have clarified in the manuscript that implementation is not a focus of the Charter tool (line 216), nor has impact been tracked or measured yet (line 257). We have also reiterated, as noted above, the scope of the paper (line 218) |
Reviewer 3 Report
The manuscript is an analysis of the state of reducing supply uncertainty of food to certain populations. The paper presents an actual analysis and proposed solutions from the simplest to the more advanced. Solutions from collaboration between government agencies and research agencies are presented, fostering the implementation of specific solutions to reduce food supply risk. The paper also addresses the aspect of food quality. Food safety consists not only in its quantitative safety, but also in its qualitative safety, having a direct impact on the health of society.
- In the methodology the authors should describe in detail their own method how they will implement their own paths and solutions to search for improvement of food safety, welfare of the society.
- Authors should present their own method, its detailed elements according to which it will be possible to achieve safety.
- According to the scheme presented in Figure 1, it is not clear what the method of improving food safety is, how the authors will assess the nutritional value of food, how the method of educating the public about nutrition will be carried out.
- In the chapter results it is expected to present own research results, while the authors in table 1 describe principles, guidelines already existing, there should be presented model solutions being the result of innovative work of the authors' team.
- The authors should emphasize more the connection between Chapter 2 Materials and Methods being an integral part of the novel approach and the results obtained in Chapter 3 Results.
Author Response
Reviewer 3 |
|
The manuscript is an analysis of the state of reducing supply uncertainty of food to certain populations. The paper presents an actual analysis and proposed solutions from the simplest to the more advanced. Solutions from collaboration between government agencies and research agencies are presented, fostering the implementation of specific solutions to reduce food supply risk. The paper also addresses the aspect of food quality. Food safety consists not only in its quantitative safety, but also in its qualitative safety, having a direct impact on the health of society |
Comments noted, thank you. While food quality is the focus rather than food safety (beyond scope of the Charter), we acknowledge its importance. |
Received additional review comments 30/5/22: |
It may be that there is a misunderstanding about the Charter - it does not deal with the topic of food safety – an area of practice governed by different policy environment and governing bodies. Therefore, food safety is outside the scope of this Charter and this manuscript.
While we have several policy actors on this author team in public health nutrition (prevention, health promotion strategy), they are not responsible for implementation on food safety (health protection policy/environmental health). It could be explored in future. |
2. Authors should present their own method, its detailed elements according to which it will be possible to achieve safety. |
We acknowledge the importance of food safety in food relief. Still, food safety is beyond scope of the current Charter. It could be explored in future. |
3. According to the scheme presented in Figure 1, it is not clear what the method of improving food safety is, how the authors will assess the nutritional value of food, how the method of educating the public about nutrition will be carried out. |
As noted above, this information is absent as it is beyond scope. Nutrition education is also not the focus of the Charter – as noted elsewhere, nutrition guidelines are in development to improve the structures and systems of food supply, rather than individual-level food choices of people experiencing food insecurity. Food education is governed by different policy context and actors at the local level. |
4. In the chapter results it is expected to present own research results, while the authors in table 1 describe principles, guidelines already existing, there should be presented model solutions being the result of innovative work of the authors' team. |
The principles for practice are generated from a synthesis of existing information, much like any systematic review would do. In this case, the consultations with practitioners refined the principles, which is the innovative aspect. We have clarified the results section accordingly, line 180-182, and 184-185. |
5. The authors should emphasize more the connection between Chapter 2 Materials and Methods being an integral part of the novel approach and the results obtained in Chapter 3 Results. |
Thanks for this suggestion, we have clarified accordingly in the results section, line 175-177. |
Reviewer 4 Report
Thank you for the opportunity to review this article about the development of a food relief charter in South Australia. The Charter answers calls from across many sectors, not just food relief, for more collaborative approaches oriented towards more holistic outcomes. I also believe it is of value to the field to have these processes documented, including in academic journals.
I think the contribution of the article could be enhanced with clearer presentation of the Charter development process. Perhaps this could be achieved by presenting it as a figure, in a similar vein to Figure 1 but focused specifically on the Charter development process. What was done, with whom, when, etc. At present, it’s difficult to distinguish the Charter development process from the overall project, which is clearly much broader in scope. The necessary information is all there, it can just be made clearer.
As the process of developing a Charter is unique because it will depend on context (i.e., local needs need to be considered and relevant stakeholders engaged), the real contribution of this article is presenting a method/model for doing so that others can draw on. The resultant outcomes (e.g., the principles derived) are important and should remain in the article, in my view, but as demonstrative of the utility of the process.
Minor points:
· -P2, line 65. I think the ABS stat that 4% of families in financial distress sought support from community organisations would be enhanced by pointing out that over one third of Australian families were in some degree of financial distress. E.g. “Government surveys in 2020 found that 38% of families reported at least one indicator of financial distress. Of these, 4% sought assistance from welfare/community organisations (including food relief), double the rate of the previous survey period”. It’s also perhaps relevant that 14% sought help from friends and family, suggesting that welfare/community organisations are something of a last resort for people and, as such, rates of access may belie or understate need.
· -I completely understand what the authors are saying with this statement “In line with this way of procuring services (distribute food) rather than social outcomes (reduce food insecurity), no sector coordination, accountability or governance mechanisms are in place” but it needs a bit of tidying. Suggestion: In line with this way of procuring services, such that the focus is on discrete support to meet an immediate need (i.e., distribute food) rather than achieving broader social outcomes (i.e., reduce food insecurity), no mechanisms to support or systematise sector coordination, accountability or governance are in place.
· -The relationship between Table 1 and Table 2 could be elaborated (i.e. how did the guiding principles from existing Charters inform the SA principles?)
· -I would refocus the discussion by bringing forward the strengths and limitations section. Typically this section is about strengths and limitations of the paper, not the process, so I think the strengths and limitations discussed would work better towards the beginning of the discussion, before the points about how each of the principles could be actioned.
· -Few typos throughout – one or two more close reads should sort them out.
Author Response
Reviewer 4 |
|
Thank you for the opportunity to review this article about the development of a food relief charter in South Australia. The Charter answers calls from across many sectors, not just food relief, for more collaborative approaches oriented towards more holistic outcomes. I also believe it is of value to the field to have these processes documented, including in academic journals |
Thank you for acknowledging the manuscript’s contribution in documenting the process of Charter development, and its value to food relief and other sectors, including academia. |
I think the contribution of the article could be enhanced with clearer presentation of the Charter development process. Perhaps this could be achieved by presenting it as a figure, in a similar vein to Figure 1 but focused specifically on the Charter development process. What was done, with whom, when, etc. At present, it’s difficult to distinguish the Charter development process from the overall project, which is clearly much broader in scope. The necessary information is all there, it can just be made clearer |
Thank you for the suggestion. We have: created a simple process diagram to distinguish the charter development steps separate to the overall partnership project on food security; and relocated text, renaming the partnership project initiation and evidence-gathering phase as ‘pre-development’ to separate it from the Charter. |
As the process of developing a Charter is unique because it will depend on context (i.e., local needs need to be considered and relevant stakeholders engaged), the real contribution of this article is presenting a method/model for doing so that others can draw on. The resultant outcomes (e.g., the principles derived) are important and should remain in the article, in my view, but as demonstrative of the utility of the process.
|
Thank you for this valuable interpretation, we agree, and have emphasised in the discussion that the key contribution of this article is presenting a method/model for doing so that others can draw on; line 227.
|
Minor points: P2, line 65. I think the ABS stat that 4% of families in financial distress sought support from community organisations would be enhanced by pointing out that over one third of Australian families were in some degree of financial distress. E.g. “Government surveys in 2020 found that 38% of families reported at least one indicator of financial distress. Of these, 4% sought assistance from welfare/community organisations (including food relief), double the rate of the previous survey period”. It’s also perhaps relevant that 14% sought help from friends and family, suggesting that welfare/community organisations are something of a last resort for people and, as such, rates of access may belie or understate need |
An excellent point, thank you; we have added to the background so as not to understate the problem – food relief is absolutely utilised as a last resort which isn’t represented in the figures. |
I completely understand what the authors are saying with this statement “In line with this way of procuring services (distribute food) rather than social outcomes (reduce food insecurity), no sector coordination, accountability or governance mechanisms are in place” but it needs a bit of tidying. Suggestion: In line with this way of procuring services, such that the focus is on discrete support to meet an immediate need (i.e., distribute food) rather than achieving broader social outcomes (i.e., reduce food insecurity), no mechanisms to support or systematise sector coordination, accountability or governance are in place |
Thank you for the rephrasing, we have edited accordingly (line 74). |
The relationship between Table 1 and Table 2 could be elaborated (i.e. how did the guiding principles from existing Charters inform the SA principles?) |
We agree, there is a lot unsaid because so much was done in iterative conversations with sector representatives to arrive at the final principles. We clarified the process in: Figure 2; line 171; and lines 190-194, to better describe this. |
-I would refocus the discussion by bringing forward the strengths and limitations section. Typically this section is about strengths and limitations of the paper, not the process, so I think the strengths and limitations discussed would work better towards the beginning of the discussion, before the points about how each of the principles could be actioned. |
A great suggestion, we have reorganised accordingly. |
Few typos throughout – one or two more close reads should sort them out. |
Noted, we have proof-read and edited. |
Round 2
Reviewer 2 Report
Thanks to the authors for providing more information. That does help to some extent. However, the overall quality of this manuscript does not change much. For a social issue such as food insecurity, to address it a charter can be easily developed, but the key challenges are to transform the charter/vision into effective actions.
The root problem of this manuscript is that there isn't much academic value in just reporting how a charter was developed. Even mentioned WA framework is quite different from the developed charter in this case.
Yet, I will change my recommendation in hoping the authors will report how the developed charter will help in addressing the food insecurity issue in the future.